# Return-to-Work Screening by Linear Discriminant Analysis of Heart Rate Variability Indices in Depressed Subjects

**DOI:** 10.3390/s21155177

**Published:** 2021-07-30

**Authors:** Toshikazu Shinba, Keizo Murotsu, Yosuke Usui, Yoshinori Andow, Hiroshi Terada, Nobutoshi Kariya, Yoshitaka Tatebayashi, Yoshiki Matsuda, Go Mugishima, Yujiro Shinba, Guanghao Sun, Takemi Matsui

**Affiliations:** 1Department of Psychiatry, Shizuoka Saiseikai General Hospital, Shizuoka 422-8527, Japan; keizo.murotsu@gmail.com; 2Autonomic Nervous System Consulting, Shizuoka 420-0839, Japan; mailthsy@icloud.com; 3Department of Psychiatry, Shizuoka Red Cross Hospital, Shizuoka 420-0853, Japan; yusui@szrc.org; 4Yoga Mental Clinic, Tokyo 158-0097, Japan; andow@rr.iij4u.or.jp; 5Aoi Clinic, Shizuoka 420-0033, Japan; aoimentalclinic@mbp.nifty.com; 6Maynds Tower Mental Clinic, Tokyo 151-00053, Japan; info@mmc-mcc.com; 7Affective Disorders Research Project, Tokyo Metropolitan Institute of Medical Science, Tokyo 156-8506, Japan; tatebayashi-ys@igakuken.or.jp (Y.T.); matsuda-ys@igakuken.or.jp (Y.M.); 8School of Human and Social Sciences, Fukuoka Prefectural University, Tagawa 825-8585, Japan; mugi@fukuoka-pu.ac.jp; 9Graduate School of Informatics and Engineering, The University of Electro-Communications, Tokyo 182-8585, Japan; guanghao.sun@uec.ac.jp; 10Graduate School of System Design, Tokyo Metropolitan University, Tokyo 191-0065, Japan; tmatsui@tmu.ac.jp

**Keywords:** depression, sick leave, return-to-work screening, heart rate variability, discriminant analysis, autonomic dysregulation

## Abstract

Using a linear discriminant analysis of heart rate variability (HRV) indices, the present study sought to verify the usefulness of autonomic measurement in major depressive disorder (MDD) patients by assessing the feasibility of their return to work after sick leave. When reinstatement was scheduled, patients’ HRV was measured using a wearable electrocardiogram device. The outcome of the reinstatement was evaluated at one month after returning to work. HRV indices including high- and low-frequency components were calculated in three conditions within a session: initial rest, mental task, and rest after task. A linear discriminant function was made using the HRV indices of 30 MDD patients from our previous study to effectively discriminate the successful reinstatement from the unsuccessful reinstatement; this was then tested on 52 patients who participated in the present study. The discriminant function showed that the sensitivity and specificity in discriminating successful from unsuccessful returns were 95.8% and 35.7%, respectively. Sensitivity is high, indicating that normal HRV is required for a successful return, and that the discriminant analysis of HRV indices is useful for return-to-work screening in MDD patients. On the other hand, specificity is low, suggesting that other factors may also affect the outcome of reinstatement.

## 1. Introduction

For workers who suffer from depression and take sick leave, the restoration of their ability to work is an important goal of treatment [1]. Restoration of work indicates not only the recovery from illness, but a return to daily life [2]. Adequate return is necessary to maintain recovery, as recurring absences due to the exacerbation of depression can be frequent and lead to deleterious effects on individuals’ future social functioning [3,4]. To assist in the judgement for restarting work, the prognostic factors related to successful and unsuccessful returns should be considered. It is reported that age, comorbid illness, severity of depression, and personality traits are associated with the outcome of reinstatement [5].

In the search for more objective prognostic factors for adequate returns to work, biological parameters have been evaluated [6]. Our previous study [7] examined the usefulness of autonomic measurement in predicting the outcome of a return to work. In that study, heart rate variability (HRV) was used to evaluate autonomic activity at the time of reinstatement, and significant differences were observed between the HRV data of the depressed patients who returned to work successfully and those who did not. The results indicated that HRV measurements are useful for evaluating the feasibility of reinstating depressive patients.

HRV has been used extensively in studies on major depressive disorder (MDD), and it is useful for understanding MDD’s pathophysiology [8]. Among various HRV indices, frequency-domain parameters are frequently utilized to analyze autonomic activity, the parameters of which include a high frequency (HF) component related to respiratory rhythm and a low frequency (LF) component related to blood pressure rhythm [9,10]. HF is known to reflect parasympathetic activity. In our previous studies [11,12], the power spectrum of heartbeat interval data was measured both at rest and during a mental task, and several HRV changes related to MDD were observed. The changes included decreased HF and increased LF/HF at rest, attenuated HF suppression during the task, and rebound-like increment of HF after the task. As such, the changes were considered state-dependent in relation to depressive symptoms.

Our study reported that these abnormal HRV profiles continue to be present at the time of return in patients whose return is unsuccessful despite their depressive symptoms having lessened [7]. The results suggest that abnormal HRV is related to a residual depressive state and can be linked to a patient’s lack of ability to perform work, as autonomic controllability is important in occupational activities and in the ability to face stressful events at work.

To further utilize the HRV findings to check the feasibility of returning to work, the use of discriminant analysis is assessed in the present study. It has been reported that linear discriminant analysis of HRV indices can differentiate MDD from a healthy control with high sensitivity and specificity [13,14], and it may be used as an objective diagnostic tool for MDD. In the present study, we extend the use of linear discriminant analysis to the HRV data of MDD patients returning to work after sick leave. If the discriminant analysis of HRV indices can differentiate successful from unsuccessful returns, HRV measurement at reinstatement in MDD patients will be an objective tool for return-to-work screening.

The present study first analyzed our previously published data [7] and made a linear discriminant using the HRV indices to effectively differentiate MDD patients who successfully returned from those did not. Then, the same discriminant was tested in new MDD patients who tried reinstatement after sick leave, and sensitivity and specificity were assessed. Discriminant function is simple and will be informative for both medical and non-medical staff who judge whether or not an MDD patient can return to work.

We also used a small wearable electrocardiogram (ECG) device for HRV measurement [7,11,12]; this allowed stable recording in various places, including the hospital and the workplace. Measurement during behavioral tasks is also possible, enabling the assessment of HRV changes in experimental settings that simulate work [15,16]. The results support the usefulness of linear discriminant analysis of HRV indices for return-to-work screening in MDD patients. The present findings are based on the patents [17,18,19].

## 2. Materials and Methods

### 2.1. Participating Patients and Study Design

This study included patients diagnosed with MDD based on the Diagnostic and Statistical Manual of Mental Disorders, Fifth Edition (DSM-5) [20], and who were treated at Shizuoka Saiseikai General Hospital, Shizuoka Red Cross Hospital, Yoga Mental Clinic, and Aoi Clinic. All patients provided written informed consent. The protocol of the study was approved by the Institutional Review Board of Shizuoka Saiseikai General Hospital. Patients were workers who took sick leave because of MDD and tried to return to their original work after their attending psychiatrists (TS, YU, YA, and HT) acknowledged the remission of MDD symptoms based on DSM-5 criteria [20]. Sick leave in the present study meant that the depressed patient took days off due to the illness. During the period of sick leave, the psychiatrists treated the patients conventionally with antidepressant medication and supportive psychotherapy. The antidepressants were selected by the attending psychiatrists. The patients consisted of two groups: those from our previously published study [7] (Group 1) and those newly enrolled consecutively in the present study (Group 2).

Group 1 consisted of 30 patients (age: 42.7 ± 12.1 years old, mean ± SD, 16 men and 14 women). The HRV data were obtained twice, once at the beginning of a patient’s sick leave and again when they returned to work [7]. The HRV data were not different between the patients in terms of successful and unsuccessful returns at the beginning of sick leave. No differences were observed for age, gender, medication dose, severity of depressive symptoms, and task performance during HRV measurement. At the reinstatement, the HRV data were significantly different between MDD patients who returned successfully and those who did not. These data were used in the present study to make a linear discriminant.

Group 2 included 52 MDD patients (age: 41.5 ± 11.5 years old, mean ± SD, 27 men and 25 women) who were newly enrolled in the present study to analyze the usefulness of this discriminant. HRV indices were measured once when their reinstatement was scheduled. HRV indices at their return were used for the discriminant function based on the Group 1 data to test the feasibility of the present method.

### 2.2. Successful and Unsuccessful Returns

One month following patients’ return to work, reinstatement was evaluated as successful or unsuccessful depending on the state of work. A reinstatement was successful when the original work was maintained. It was unsuccessful when the content of work changed, the patient took another sick leave, or the patient resigned from work. Work conditions were verified using the information obtained from the affiliated company.

The observation of work conditions for a longer interval from the return to work should be informative for accurately evaluating the outcome. However, the effects of various occupational, social, and personal events could more often affect the return outcome if a longer interval between the return and the work condition evaluation was employed, making an analysis of the significance of autonomic data at the return for the outcome more complex. Thus, we used 1 month as an evaluation interval to examine the feasibility of the present method.

### 2.3. Heart Rate Variability Measurement

The methods of HRV measurement were the same as those found in our previous study [7]. After an adaptation period of at least 5 min, the patient was seated on a chair with electrocardiogram (ECG) electrodes of a wireless amplifier (40.0 × 35.0 × 7.2 mm, 12 g, RF-ECG2, GM3, Tokyo, Japan) attached to their chest. ECG was recorded conventionally with a gain of 10,000 and a time constant of 0.1 s, and the data were stored in a computer with a sampling frequency of 200 Hz. R peaks were used to make the R-R interval trend data. R-R intervals between the range of 273 and 1500 ms were used for analysis to exclude paroxysmal heart beats. When an R-R interval was omitted, it was replaced by the average of the preceding and following intervals. The R-R interval trend data were resampled using the mean heart rate (HR), and their fluctuation was analyzed using the maximum entropy method (MemCalc, GMS, Tokyo, Japan) to determine the LF and HF components of the spectrum every 2 s by integrating the power to corresponding frequency intervals (0.04–0.15 Hz for LF; 0.15–0.4 Hz for HF) [21] for the preceding 30 s period. R-R intervals were also converted to HR (/min). The maximum entropy method was selected for the power spectrum analysis because it has been successfully applied to trend data with a minimum duration of 30 s; moreover, it is useful for studies incorporating measurements of multiple behavioral states [22].

Respiration was monitored and its frequency was confirmed to be within the range of 0.15–0.4 Hz in each participant [23]. When the respiratory frequency exceeded the aforementioned range, patients were asked to modulate their rate of breathing. Then, the measurement was repeated. Breath rate modulation was instructed only at the start of the measurement when the study staff noticed that the respiratory frequency exceeded the range, because some patients intended to slow their breathing rate. After the instruction to breathe at a usual pace, all the patients could continue their breathing within the normal frequency range.

### 2.4. Experimental Protocol

ECG was recorded in three different conditions (AMAS, GM3, Tokyo, Japan). First, the patients were instructed to relax as much as possible on the chair for approximately 60 s (initial rest; rest). Then, they were engaged in a random number generation task [24] for 100 s (mental task; task). After the task, ECG was recorded for another 60 s while the patient was in a relaxed state (rest after task; after). HF, LF, LF/HF, and HR were averaged in the interval from 30 s after the onset to the end of each condition to exclude any data at the beginning of each new period that might have reflected the previous condition [11,12,22]. The data in the task and after conditions were expressed as ratios to the data in the rest condition (task/rest and after/rest).

In the random number generation task, the patients were instructed to orally generate a random series of 100 digits using the numbers 0 through 9 at a rate of 1 Hz. The generation rate was indicated by a metronome click sound. They were requested to concentrate on this task as much as possible [24].

### 2.5. Discriminant Analysis and Statistics

The HRV indices obtained in the previous study (Group 1) served as the data for making the discriminant function. Discriminant analysis was performed to make a linear equation composed of the HRV indices multiplied by coefficients and a constant (Equation (2)). Values of the coefficients were set so that the discriminant function could significantly differentiate the Group 1 patients with a successful return from those with an unsuccessful return by making the discriminating point zero. The score of the discriminant function (discriminant score) is calculated and is positive when reinstatement is successful and negative when unsuccessful (StatMate V, ATMS, Chiba, Japan). A detailed flow of the calculation is found in the reference for this software [25], and is summarized in the Results section.

Sensitivity and specificity in discriminating successful from unsuccessful returns using the discriminant scores were calculated in both Group 1 and Group 2. Further, 95% confidence interval (CIs) for sensitivity and specificity were calculated using Wilson method [26]. HRV indices, discriminant scores, and age were compared between the true and false positive patients in Group 2 using the Mann–Whitney U test.

## 3. Results

### 3.1. Linear Discriminant Function Made from the Previous Data

The discriminant function included HF, LF, and LF/HF indices at rest, task, and after, as shown below (Equation (2)). Linear discriminant analysis tested whether successful (n = 19) and unsuccessful patient data (n = 11) of the nine-dimensional variable in Group 1 could be separated by a line (linear discriminant function) expressed as Equation (2). The step-by-step procedures are summarized below [25].

(1)Successful and unsuccessful return data matrices are obtained.(2)Linear discriminant function is determined as follows.
i.The sum of products of deviations in total return data is calculated by adding the sum of products of deviations in the successful return data and that in the unsuccessful return data.ii.The variation-covariation matrix in total return data is calculated using the sum of produces of deviations in total return data.iii.The mean difference matrix for successful and unsuccessful data is calculated.iv.The coefficients (a to i) are determined using the product of the inverse matrix of the variation-covariation matrix in total and the mean difference matrix.v.Then, the following function (Equation (1)) is obtained.


Y = a HF[Rest] + b HF[Task/Rest] + c HF[After/Rest]+ d LF[Rest] + e LF[Task/Rest] + f LF[After/Rest]+ g LF/HF[Rest] + h LF/HF[Task/Rest] + i LF/HF[After/Rest](1)

(3)Mahalanobis’ generalized distance is obtained by the sum of produces of coefficients and mean differences.(4)F test is used to test whether the discriminant function can significantly differentiate the data in a successful return and that in an unsuccessful return. Mahalanobis’ generalized distance is used for F_0_.(5)Discrimination point is determined as the center of scores obtained by the function for successful return data and for unsuccessful return data.(6)Discriminant function is determined using the discrimination point as a constant in the equation as follows.

discriminant score = a HF[Rest] + b HF[Task/Rest] + c HF[After/Rest] + d LF[Rest] + e LF[Task/Rest] + f LF[After/Rest] + g LF/HF[Rest] + h LF/HF[Task/Rest] + i LF/HF[After/Rest] − discrimination point (2)

In the present study, discriminant score >0 indicates that the data belong to a successful return, and discriminant score <0 indicates that the data belong to an unsuccessful return, based on the following discriminant function.

Mahalanobis’ generalized distance was 13.36, and the *p*-value was less than 0.001 for the present equation, indicating that the discrimination was possible for Group 1 data (StatMate V). The discriminant function made for the Group 1 data was used for the Group 2 data.

LF/HF was included because use of LF/HF increased sensitivity and specificity. Task/rest and after/rest ratios were also employed because they were found to be abnormal in MDD in our previous studies [7,11,12,27], and were evaluated as reactivity of HRV, which is useful in analyzing the autonomic activity. HR indices were not used because the inclusion of HR data did not improve sensitivity and specificity. Other different sets of measures were tried, but the sensitivity and specificity of the Group 1 data were lower than the present set including nine measures. The sensitivity and specificity to discriminate a successful return from an unsuccessful return in this group of patients reached 100% (CI: 83.2–100%) and 90.9% (CI: 62.3–98.4%), respectively (Table 1, Group 1). False negative and false positive rates were 0% and 9.1%, respectively. The discriminant scores of the Group 1 patients with successful and unsuccessful returns are presented in Figure 1 (Group 1).

### 3.2. Application of Linear Discriminant Function to New MDD Patients

In 52 Group 2 patients, HRV was measured at the time when the reinstatement was scheduled, and discriminant score was calculated by the discriminant function shown above. The cutoff point was set at zero in the discriminant function determined with Group 1 patients, and was also used for Group 2 patients. Forty-one patients exhibited positive scores, while 11 showed negative scores. One month following their return to work, 24 patients continued their original work, while 28 failed to continue. The number of successful and unsuccessful returns with positive or negative scores is summarized in Table 1 (Group 2). Using the discriminant function, the sensitivity and specificity to discriminate a successful return from an unsuccessful return in this group of MDD patients were 95.8% (CI: 79.8–99.3%) and 35.7% (CI: 20.7–54.2%), respectively. False negative and false positive rates were 4.2% and 64.3%, respectively. Figure 1 (Group 2) shows the distribution of discriminant scores for both successful and unsuccessful returns. The scores in the successful returners are mostly positive. On the other hand, the scores in the unsuccessful returners exhibited a wide variation.

Table 2 shows the averaged data of HRV indices in the patients with successful and unsuccessful return whose discriminant scores were positive. The statistical differences between the data of successful and unsuccessful returns were checked using Mann–Whitney U test. Significant differences were observed for HF rest score, HF task/rest ratio, LF/HF rest score, and LF/HF task/rest ratio (*p* < 0.05). The mean discriminant scores were also different between the successful returners (7.37 ± 4.05) and the unsuccessful returners (4.09 ± 2.88, *p* = 0.006). As for age, the successful returners (44.7 ± 10.2 years) were older than the unsuccessful ones (36.0 ± 12.2, *p* = 0.013).

## 4. Discussion

The present study indicates that the linear discriminant analysis of HRV indices is useful in judging the feasibility of reinstatement, and has advanced our previous findings [7]. The discriminant function was based on the previous data of 30 patients. The same discriminant function was applied to 52 patients newly enrolled in the study, and the sensitivity to discriminate a successful return from an unsuccessful return reached as high as 95.8%. It is indicated that normal HRV is necessary to make a successful return. High sensitivity was observed both in the group of patients whose data were used for the discriminant function and in the group of new patients. This supports the reliability of the present discriminant method (Table 1, Group 2).

The present discriminant is made on a database of 30 MDD patients whose age, gender, anti-depressant dosage, symptom severity, randomness score of task performance, and duration of sick leave did not differ between successful and unsuccessful returns [7], thus diminishing the possible effects of these parameters. It is suggested that this discriminant function should reflect the relation of HRV to the outcome of reinstatement.

The present study measured not only the baseline autonomic activity during the initial resting state, but also the autonomic reactivity during task load, as well as during the resting state after the task [11,27]. Depending on behavioral conditions, adequate modulation of autonomic function should be associated with the outcome of reinstatement because a modification of sympathetic and parasympathetic balance is important to manage the stressful situations in a daily work environment. We normalized the data at ‘task’ and those at ‘after’ because previous works have indicated that normalization can reveal the autonomic dysregulation of depression, which is not observed when only the raw data are used [11,12,27]. The task/rest and after/rest ratios signify the reactivity of the autonomic system to work load, and are informative with respect to evaluation of the autonomic regulation in depressed patients. Scale differences exist, but the present discriminant function has worked well to distinguish the successful and unsuccessful returns.

Our previous study [7] showed that the HRV indices are state-dependent and can change by treatment. Amelioration of autonomic dysfunction revealed by HRV analysis could be related to psychopathological improvement, and it can serve as a biological marker by reflecting improvements in depressive illness. The present study further advanced the utility of HRV indices using discriminant scores to simplify the data output.

On the other hand, the specificity of the present discriminant to differentiate successful from unsuccessful returns was low (35.7%) in the newly enrolled patients. More than half of the patients predicted to return successfully did not actually make successful reinstatements. Normal autonomic function is necessary, but it is not enough to accomplish a successful return. Personal factors including physical conditions and personality traits, as well as external factors such as inter-personal relationships at work and daily life, should be important in enabling a successful return [5]. Future studies incorporating multiple parameters other than HRV are warranted so that we might increase the specificity of the discriminant analysis.

The present data further indicated that some indices show differences between the patients who were predicted to return successfully and did so and those who were predicted to return successfully but did not. The unsuccessful returners were younger than the successful returners. The age was not statistically different when all Group 2 patients were analyzed (successful: 44.5 ± 10.0 years old, unsuccessful: 39.0 ± 12.2, *p* > 0.05), but was significantly different when the patients with positive discriminant scores were analyzed. The difference in age may be related to differences in occupational and daily life conditions. Their discriminant scores were also different, as were some HRV indices. In comparison with the successful returners, the unsuccessful returners who were predicted to return successfully exhibited a lower HF rest score, a higher HF task/rest ratio, a higher LF/HF rest score, a lower LF/HF task/rest ratio, and a positive but lower discriminant score. These HRV profiles in the unsuccessful returners are generally common with the HRV profiles of MDD. That said, the raw scores of HRV indices in unsuccessful returns with successful predictions are not abnormal, as evidenced in the data from our previous publication [7,11,12,27]. It is important to carefully follow the patients with positive but low discriminant scores after their return to work.

In the study on the Group 1 patients [7], psychological assessments including self-rating depression scale (SDS) and state-trait anxiety inventory (STAI) were used for analysis. These psychological scores improved significantly in both the successful and unsuccessful returners at the time of reinstatement. However, the HRV measures became normal only in the successful returners. The attending psychiatrists acknowledged the symptom remission and evaluated that the reinstatement was possible for the unsuccessful returners, but the HRV data indicated the presence of risk regarding the return to work. We thus used only HRV measures for the present discriminant function, but psychological indices are informative and future studies incorporating them into the discriminant function should be interesting.

The present study has some limitations. The sample size was small and the outcome of return was evaluated only 1 month after a return to work. Future studies with a larger sample size and with a longer period for evaluation of the return are warranted. It will be interesting to update the discriminant function by analyzing the HRV data, including both Group 1 and Group 2 patients, and to try to analyze this function on new patients in a future study. Incorporation of other biological and psychological indices, including skin conductance and age, as well as environmental indices such as present work condition and employment length, into the discriminant function would also be promising.

Future trials are warranted to continue testing the present discriminant function in various clinical and occupational situations in which a return to work is judged in relation to MDD patients after a period of sick leave. At present, few objective measures are available to determine the appropriate time for reinstatement [6], and the HRV measurement can provide useful information regarding the feasibility of a return to work. By incorporating the mental task into an autonomic measurement, the present method could be useful for simulating the reality of the workplace. The present study also employed a small wearable device to measure ECG for HRV analyses. The R peaks are clear and make the interval trend data accurate. The equipment is small and portable, enabling the simple but medically reliable system for return-to-work screening in depressed patients.

## Figures and Tables

**Figure 1 sensors-21-05177-f001:**
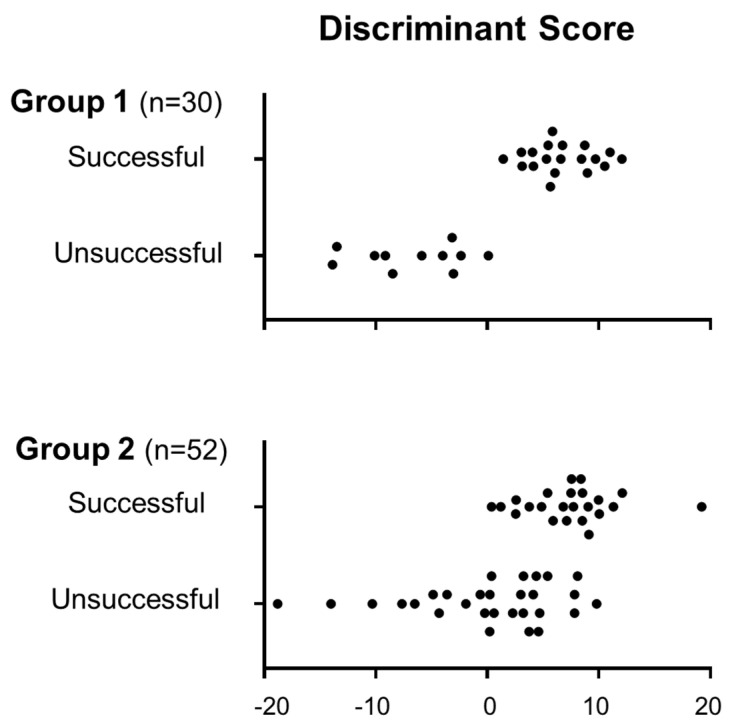
Distribution of discriminant scores in Group 1 and Group 2 patients with successful returns (successful) and unsuccessful returns (unsuccessful). Each filled circle indicates individual data.

**Table 1 sensors-21-05177-t001:** Frequency Distribution of Discriminant Scores in Successful and Unsuccessful Returns.

Group 1 (n = 30)				
		Successful	Unsuccessful	Total
	Positive	19	1	20
	Negative	0	10	10
	Total	19	11	
		*sensitivity*	*specificity*	
		100%	90.9%	
**Group 2** **(n = 52)**				
		Successful	Unsuccessful	Total
	Positive	23	18	41
	Negative	1	10	11
	Total	24	28	
		*sensitivity*	*specificity*	
		95.8%	35.7%	

**Table 2 sensors-21-05177-t002:** Age, Discriminant Score, Heart Rate, and Heart Rate Variability Indices in Successful and Unsuccessful Returns with Positive Discriminant Scores.

**Age**	years					
Successful	44.7	(10.2)				
Unsuccessful	36.0	(12.2)				
*p*	*0.013*					
**Discriminant Score**						
Successful	7.37	(4.05)				
Unsuccessful	4.09	(2.88)				
*p*	*0.006*					
	**Rest**		**Task/Rest**		**After/Rest**	
**Heart Rate**	/min					
Successful	76.9	(10.0)	1.04	(0.05)	0.98	(0.04)
Unsuccessful	80.3	(16.0)	1.04	(0.04)	1.00	(0.06)
*p*						
**HF**	ms^2^					
Successful	250.8	(240.5)	0.36	(0.23)	1.64	(0.81)
Unsuccessful	109.2	(102.3)	0.59	(0.30)	1.56	(0.81)
*p*	*0.034*		*0.006*			
**LF/HF**						
Successful	1.19	(0.91)	5.67	(5.00)	2.42	(3.65)
Unsuccessful	3.04	(3.17)	2.32	(1.90)	1.78	(1.45)
*p*	*0.003*		*0.001*			
**LF**	ms^2^					
Successful	274.7	(360.1)	1.74	(1.18)	3.94	(7.88)
Unsuccessful	251.7	(215.4)	1.25	(0.96)	2.18	(1.47)
*p*						

The data are expressed as mean (S.D.). The unit of the data is presented above the data when applicable. *p*: *p*-value (Mann-Whitney U test) less than 0.05 below the data indicates that the difference between Successful and Unsuccessful groups is statistically significant. HF: high frequency component, LF: low frequency component, LF/HF: ratio of LF to HF. See the detailed descriptions on Rest, Task/Rest, and After/Rest in the text.

## Data Availability

The data that support the findings of this study are available on request from the corresponding author. The data are not publicly available due to privacy and ethical restrictions.

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
