# Peer review of "Return-to-Work Screening by Linear Discriminant Analysis of Heart Rate Variability Indices in Depressed Subjects"

_sensors, 2021, doi:10.3390/s21155177_

Round 1
Reviewer 1 Report
This is an outstanding contribution and builds on the authors’’ prior work. Prediction of treatment stability is a significant and continuing problem in clinical practice. The work presented here identifies a promising avenue for future work. Requests for calcification follow
To be absolutely sure of my understanding: Coefficients of Equation 1 were determined with Group 1 data and were used to classify Group 2 patients. Was the cutoff score determined with Group 1 patients used to classify Group 2 patients?
A clarification of language in Section 2.1 would be helpful
Line 115. “The HRV data were not different between the patients in terms of successful and unsuccessful returns at the beginning of sick leave.”
Line 118. “The HRV data were significantly different between MDD patients who returned successfully and those who did not.”
Am I correct in understanding that the sentence beginning at Line 118 refers to measures obtained when reinstatement was scheduled?
Were standardized depression assessments (for example, the Beck Depression Inventory or the Hamilton Depression Rating Scale) used? If they were, what quantitative relationships were observed between these assessments and HRV measures? Specifically, were physiological measures more predictive of reinstatement success than clinical assessments?
Regression Details
Bootstrapped confidence intervals for sensitivity and specificity would be helpful.
Three types of HRV measures were use: Rest, Task/Rest and After/Rest. As shown in Table 2, these measures have very different magnitude ranges. Scale differences of this magnitude can cause problems in regression calculations. Why was this normalization employed? Were calculations performed with unnormalized Task and After data?
Why were all nine measures used? Was any consideration given to employing a quantitative feature selection procedure?
How was the cutoff score determined (for example maximization of the Youden Index)?
Line 287. “The unsuccessful returners were younger than the successful returners.”
This would seem to argue for incorporation of age into the regression model. Length of time in their pre-hospitalization employment might also be useful.
Author Response
Dear reviewer 1
Thank you very much for evaluating our work. We are planning future researches promoting HRV utilization in psychiatric practices. Your comment, “This is an outstanding contribution and builds on the authors’’ prior work. Prediction of treatment stability is a significant and continuing problem in clinical practice. The work presented here identifies a promising avenue for future work. “, encourages us a lot. We modified the manuscript according to your comments and suggestions. We hope that the manuscript is sufficiently improved. Below are the list of your comments and our responses.
Sincerely yours,
Toshikazu Shinba
Comment 1:
To be absolutely sure of my understanding: Coefficients of Equation 1 were determined with Group 1 data and were used to classify Group 2 patients. Was the cutoff score determined with Group 1 patients used to classify Group 2 patients?
Response:
Thank you for the important question. Yes, the coefficients of the variables in the discriminant function were determined using the mean difference matrix and the variance-covariance matrix of the HRV measures of Group 1 patients. A constant was introduced to the discriminant function to make the cutoff point at zero. The discriminant function made for the Group 1 data was also used for the Group 2 data (line 210, 247).
Comment 2:
A clarification of language in Section 2.1 would be helpful
Line 115. “The HRV data were not different between the patients in terms of successful and unsuccessful returns at the beginning of sick leave.”
Line 118. “The HRV data were significantly different between MDD patients who returned successfully and those who did not.”
Am I correct in understanding that the sentence beginning at Line 118 refers to measures obtained when reinstatement was scheduled?
Response:
Yes, you are right. Thank you for the valuable comment. We clarified that point in the manuscript (line 121).
Comment 3:
Were standardized depression assessments (for example, the Beck Depression Inventory or the Hamilton Depression Rating Scale) used? If they were, what quantitative relationships were observed between these assessments and HRV measures? Specifically, were physiological measures more predictive of reinstatement success than clinical assessments?
Response:
Thank you for the important comment. In the study on the Group 1 patients, psychological assessments including self-rating depression scale (SDS) and State-trait anxiety inventory (STAI) were used for analysis. These psychological scores improved significantly in both the successful and unsuccessful group patients. But the HRV measures got normal only in the successful group patients. We therefore incorporated only HRV measures for the present discriminative function. But future researches incorporating psychological indices to the discriminant function should be interesting. These descriptions were added to the discussion section of the manuscript (line 329).
Comment 4:
Regression Details
Bootstrapped confidence intervals for sensitivity and specificity would be helpful.
Response:
Confidence interval for sensitivity and specificity were calculated by Wilson method and presented in the result section (lines 201, 225, 255).
Comment 5:
Three types of HRV measures were used: Rest, Task/Rest and After/Rest. As shown in Table 2, these measures have very different magnitude ranges. Scale differences of this magnitude can cause problems in regression calculations. Why was this normalization employed? Were calculations performed with unnormalized Task and After data?
Response:
Thank you for the valuable question. We normalized the data at Task and that at After because previous work has indicated that normalization can reveal the autonomic dysregulation of depression, which is not observed when only the raw data are used. The Task/Rest and After/Rest ratios signify the reactivity of the autonomic system to work load, and are very informative with respect to evaluation of the autonomic regulation in depressed patients. The scale differences of magnitude exist but the present discriminant equation worked well to distinguish the Successful and Unsuccessful group. We added these points to the manuscript (lines 219, 292).
Comment 6:
Why were all nine measures used? Was any consideration given to employing a quantitative feature selection procedure?
Response:
Thank you for the question. We tried different sets of measures but the sensitivity and specificity for the Group 1 data were lower than the present set including nine measures. We clarified this description in the manuscript (line 220).
Comment 7:
How was the cutoff score determined (for example maximization of the Youden Index)?
Response:
Thank you for the comments. The coefficients of the variables were determined using the mean difference matrix and variance-covariance matrix of the HRV measures of Group 1 patients. A constant was introduced to the function to make the cutoff point at zero (Takahashi, 2014). The discriminant function made for the Group 1 data was also used for the Group 2 data. The description was clarified in the manuscript (line 210).
Comment 8:
Line 287. “The unsuccessful returners were younger than the successful returners.”
This would seem to argue for incorporation of age into the regression model. Length of time in their pre-hospitalization employment might also be useful.
Response:
Thank you for your important comment. The age was not statistically different when all patients were analyzed (success 44.5+/-10.0, unsuccess 39.0+/-12.2, p>0.05), but was significantly different when only the patients with positive Z-scores were analyzed. These points were clarified in the manuscript. But the age together with length of employment are important and could be useful in future studies. The description was added to the manuscript (line 316, 345).
Reviewer 2 Report
This study intended to verify the usefulness of autonomic measurement in major depressive disorder (MDD) patients by assessing the feasibility of their return to work after sick leave by using a linear discriminant analysis of heart rate variability (HRV) indices. HRV indices including high- and low-frequency components were calculated in three conditions within a session: initial rest (Rest), mental task (Task), and rest after task (After). The authors concluded that the sensitivity and specificity of Z-scores in discriminating successful from unsuccessful returns were 95.8% and 34.5%, respectively. The high sensitivity indicated that normal HRV is required for a successful return, and that the discriminant analysis of HRV indices is useful for return-to-work screening in MDD patients. The low specificity suggested that other factors may also affect the outcome of reinstatement. This is an interesting study. However, several issues need to be addressed by the authors.
Major issue
Is discriminant function the same as the Z scores in this study? The Z scores defined in this study contained 3 kinds of HRV indices: Rest, Task/Rest, and After/Rest; each kind of HRV indices contained 3 indices: HF, LF and LF/HF. The definition of Z score is not based on rational and logical argument; rather, it seems to come from nowhere. What’s the physiological meaning of the Z scores? How are the coefficients and constant in the Z scores determined?
Minor issues
- What is meant by “sick leave”? This study was carried out on MDD patients who were already sick before, during and after the study. Then what is the meaning of “sick leave” in the sick patients?
- Line 162-164. “When the respiratory frequency exceeded the aforementioned range, patients were asked to modulate their rate of breathing. Then, the measurement was repeated.” From this statement, it seems that the study subject was monitored and the recordings were manipulated by the study staff, instead of non-biased real data of the patients. Is it suitable to do this manipulation?
- Line 195. Student t test was used to compare HRV indices, Z-scores, and age between the true and false positive patients in Group 2. Have the authors checked the normality of the data distribution of the parameters used in this study?
- Line 233. The sensitivity and specificity to discriminate a successful return from an unsuccessful return in this group of MDD patients were shown. How about the false negative and false positive rate of discriminating successful return from an unsuccessful return?
Author Response
Dear reviewer 2
Thank you for your valuable comments. Following your comment, “This is an interesting study. However, several issues need to be addressed by the authors.”, we modified the manuscript. We believe that the manuscript is now sufficiently improved. Below are the list of your comments and our responses.
Sincerely yours,
Toshikazu Shinba
Comment 1:
Major issue
Is discriminant function the same as the Z scores in this study? The Z scores defined in this study contained 3 kinds of HRV indices: Rest, Task/Rest, and After/Rest; each kind of HRV indices contained 3 indices: HF, LF and LF/HF. The definition of Z score is not based on rational and logical argument; rather, it seems to come from nowhere. What’s the physiological meaning of the Z scores? How are the coefficients and constant in the Z scores determined?
Response:
Yes, Z-score is the solution of discriminant function. The coefficients of the variables in the discriminant function were determined using the mean difference matrix and variance-covariance matrix of the HRV measures of Group 1 patients (Takahashi, 2014). A constant was introduced to the function to make the cutoff point at zero. The discriminant function made for the Group 1 data was also used for the Group 2 data. Detailed description was added to the manuscript (line 210).
Comment 2:
Minor issues
What is meant by “sick leave”? This study was carried out on MDD patients who were already sick before, during and after the study. Then what is the meaning of “sick leave” in the sick patients?
Response:
Thank you for your important comment. ‘Sick leave’ in the present study meant that the depressed patient took days off due to the illness. We added the description to the manuscript (line 109).
Comment 3:
Line 162-164. “When the respiratory frequency exceeded the aforementioned range, patients were asked to modulate their rate of breathing. Then, the measurement was repeated.” From this statement, it seems that the study subject was monitored and the recordings were manipulated by the study staff, instead of non-biased real data of the patients. Is it suitable to do this manipulation?
Response:
Thank you for your important question. Breath rate modulation was instructed only at the start of the measurement when the study staff noticed that the respiratory frequency exceeded the range, because some patients intended to slow their breathing rate. After the instruction to breath in a usual pace, all the patients could continue their breathing within the normal frequency range. The description was added to the manuscript (line 168).
Comment 4:
Line 195. Student t test was used to compare HRV indices, Z-scores, and age between the true and false positive patients in Group 2. Have the authors checked the normality of the data distribution of the parameters used in this study?
Response:
Thank you for the valuable comments. We checked the normality of the data distribution and found out that some do not show normal distribution. We analyzed all data by nonparametric Man-Whitney U test. And the results are the same as those with t-test. The statistical data was changed to that by Mann-Whitney U test. P-values were renewed in the Table 2 and in the manuscript (line 204, 265).
Comment 5:
Line 233. The sensitivity and specificity to discriminate a successful return from an unsuccessful return in this group of MDD patients were shown. How about the false negative and false positive rate of discriminating successful return from an unsuccessful return?
Response:
Following your comment, we introduced the description of false negative and false positive rate and discussed the results (line 226, 255)
Round 2
Reviewer 2 Report
Though some sentences have been added to improve the description of Z-score, it is still not clear how do they obtain their Z-score.
- The Z-score in the literature is given by Z = (x – mean of x)/(standard deviation of x). Is the Z-score of the authors derived from this Z-score in the literature? If the answer is yes, the authors are advised to present their derivation of their Z-score from the Z-score in the literature step by step. If the answer is no, why do they use the name “Z-score” to confuse the readers?
- The authors state in the revised manuscript that “Z-score is the solution of this equation (discriminant function including HRV variables?). The coefficients of the variables in the discriminant function were determined using the mean difference matrix and the variance-covariance matrix of the HRV measures of Group 1 patients.” It is not clear how they obtain their Z-score through the mean difference matrix and the variance-covariance matrix of the HRV measures of Group 1 patients. The authors are advised to make it clear how they get their Z-score by giving the details of derivation so that other researchers can follow their research and verify their findings.
- How are the coefficients in the Z-score determined?
